# Diffusing wave paradox of phototactic particles in traveling light pulses

Celia Lozano [1] & Clemens Bechinger [1]

Cells navigate through complex surroundings by following cues from their environment. A prominent example is Dictyostelium, which is directed by chemotaxis towards regions with higher concentrations. In the presence of traveling chemical waves, however, amoebae migrate counter to the running wave. Such behavior, referred to as diffusing wave paradox, suggests the existence of adaptation and directional memory. Here we experimentally investigate the response of phototactic self-propelled microparticles to traveling light-pulses. Despite their entirely memory-less (i.e., strictly local) response to the environment, we observe the same phenomenological behavior, i.e., particle motion counter to the pulse direction. Our findings are supported by a minimal model which considers active particle reorientations within local light gradients. The complex and robust behavior of synthetic active particles to spatially and temporally varying stimuli enables new strategies for achieving collective behavior and can be used for the design of micro-robotic systems with limited signal-processing capabilities.

[1] Fachbereich Physik, Universität Konstanz, D-78457 Konstanz, Germany. Correspondence and requests for materials should be addressed to C.B. (email: clemens.bechinger@uni-konstanz.de)

Collective behavior of microorganisms is often achieved by the emission and detection of extracellular signaling molecules which regulates their motility[1–3]. One of the most intensively studied examples of such biochemical communication are amoebae, e.g. *Dictyostelium discoideum*, which can spontaneously reorganize from a dispersed population into a multicellular macroscopic organism[4,5]. This transition is triggered by few leader cells periodically emitting symmetric pulses of cyclic adenosine 3′,5′-monophosphate (cAMP), which travel as dissipation-free waves by a self-enhancing cell-to-cell relay over large distances[6]. When *Dictyostelium* amoebae are hit by cAMP-waves, they move counter to the wave traveling direction, which eventually leads to the formation of dense aggregates[7].

Such behavior is an integral part of their survival strategy in nutrient-deprived environments. Interestingly enough, in static cAMP gradients, *Dictyostelium* exhibit positive chemotactic behavior, i.e. it moves towards larger chemical concentrations[8,9]. Therefore, one expects them to follow the concentration maximum which should hinder their aggregation[10,11]. This seemingly inconsistent behavior, which has been recently also confirmed with motile bacteria[12], is usually referred to as *chemotactic wave paradox*[5,13]. Only recently, it has been demonstrated that it results from the finite adaptation time of the organisms to variations of the surrounding chemical concentration[11,14–16]. Such time-delayed response to spatio-temporal stimuli leads to a slightly different motional response of amoebae to the front and back of entirely symmetric cAMP waves[17]. This explains their aggregation into multi-cellular collectives, but also their astonishingly effective migration within complex cellular tissues[18] and artificial mazes[11,19].

The directed motion induced by spatio-temporal cues would be also attractive for active particles (APs), i.e. the synthetic counterpart of living microorganisms. Such systems hold potential as microrobots to carry and sense, e.g. drugs in complex environments[20–23]. Similar to motile cells, APs can harvest energy from their environment and convert it into locomotion[23]. In addition, they can be spatially directed by chemical[24–26], optical[27,28], flow[29,30], or gravitational[31,32] fields, and thus resemble basic cellular behavior. Contrary to microorganisms, the simple structure of APs does neither allow for intricate signal processing nor a time-delayed response to external stimuli[33]. Instead, APs respond strictly local to their environment[27]. Given these limitations compared to living systems, it is surprising that numerical simulations suggest the principle possibility of AP motion against and along a traveling pulse[34–36].

Here, we experimentally study the response of phototactic microparticles to slowly traveling optical light pulses, which are created by a scanned elongated focus of a laser beam. By analyzing the particles' translational and orientational motions, we demonstrate that APs indeed move either along or counter to a propagating optical pulse depending on its velocity and width. In addition to single pulses, we also investigated the particle response to periodic pulse trains. With increasing time interval between consecutive pulses, directional particle motion decreases, which is due to the diffusive decorrelation of particle orientations. Since the orientational decorrelation time strongly depends on the particle size, this effect can be promoted as a sorting mechanism, which allows directing particles of different sizes into opposite directions. Our results are supported by numerical simulations which yield good agreement with the experimental data.

## Results

**Experimental characterization of the aligning torque.** APs are fabricated from colloidal spheres with diameter $\sigma = 3.25\,\mu m$, which are half-coated by carbon cap with 50 nm thickness. When suspended in a critical mixture of water–2,6-lutidine and homogeneously illuminated with laser light, they begin to self-propel. The magnitude and directionality of the propulsion velocity $v_p$ can be controlled by the illuminating laser intensity $I$ (further details are provided in the section "Methods")[37,38]. Below and above a threshold intensity $I_r$ particles propel with the carbon cap in the back and the front, respectively. As a result of Brownian rotational motion, the direction of the propulsion fluctuates on a time scale $1/D_r \sim 5\,s$ set by the inverse rotational diffusion coefficient $D_r$. Under homogeneous light illumination, the APs perform an isotropic persistent random walk which is confined to two dimensions due to gravity and hydrodynamic interactions with the walls of the sample cell[37,39]. In the presence of a light gradient $\nabla I$, however, the APs motion is no longer isotropic but exhibits a pronounced phototactic behavior. Depending on whether the illumination intensity is below or above $I_r$, APs will propel opposite or along $\nabla I$ ("Methods")[27,37].

All measurements presented in the following were carried out for illumination intensities $I > I_r$, i.e. the particles are propelling with the cap ahead and are aligned towards increasing light intensity (positive phototaxis) ("Methods"). This is demonstrated in Fig.1a, b, where we plotted the trajectory of an AP within a one-dimensional triangular light profile. The intensity pattern is created by a scanned line-shaped laser beam, whose scanning motion is synchronized with an electro-optical modulator, the latter modulating the laser intensity ("Methods"). For $I > I_r$, the propulsion velocity $v_p$ depends almost linearly on $I$ ("Methods"), therefore it increases towards the intensity maximum (see color code of the trajectory). Because the particle reorientation

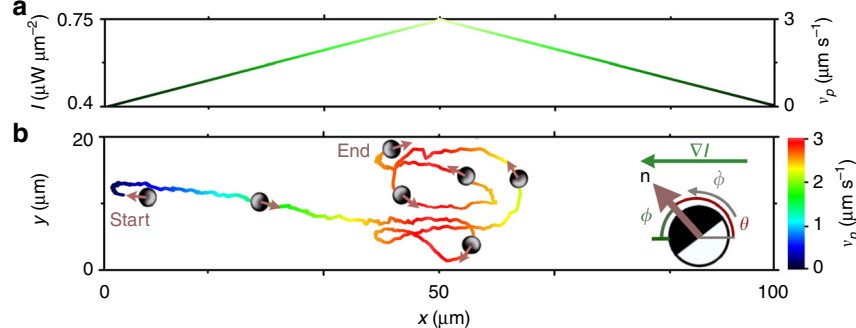

**Fig. 1** Positive phototaxis. **a** One-dimensional triangular light intensity profile (left axis) and corresponding propulsion velocity $v_p$ (right axis). **b** Measured trajectory of a positive photoacting AP in the light gradient with its propulsion velocity $v_p$ labeled in color. (Inset) Sketch of an AP in a light gradient $\nabla I$. The fluid's flow field around the particle becomes axially asymmetric relative to the particle's orientation **n**, which results in a torque which aligns the AP parallel to $\nabla I$[37]. $\phi$ is the angle between $\nabla I$ and the particle orientational vector **n**. The angle $\theta$ describes the particle orientation **n** relative to the positive axis

dynamics is not instantaneous but limited by viscous friction, this leads to visible overshoots across the intensity maximum before the AP's orientation reverses (Fig.1b). As a result, it becomes effectively localized (motility trap) near the intensity maximum where it performs an almost periodic back-and-forth motion (Supplementary Movie 1).

The aligning torque of a positively phototacting AP within a gradient $\nabla I$ is given by[27,40,41]

$$M \propto -\nabla I \times \mathbf{n} \qquad (1)$$

The corresponding reorientation dynamics can be expressed by the angular velocity.

$$\dot{\phi} = \omega_{\max} \sin \phi, \qquad (2)$$

where the amplitude $\omega_{\max}$ grows with increasing $\nabla I$ but eventually saturates[27] ("Methods"). In our experiments, all light gradients were above this saturation threshold, which results in a constant particle reorientation time $\tau = 3.3 \pm 0.75$ s being much shorter than $1/D_r$ (see the section "Methods" for further details).

**Particle response to traveling light pulses**. Apart from oscillations around the intensity maximum, no drift motion of APs is observed in a static triangular light profile. When interacting with a traveling light pulse (Fig. 2a), however, they are translated during each encounter with the pulse. Each pulse is characterized by its traveling velocity $u$, its width $w$ and the light pulse amplitude $I^{\max}$, the latter being kept constant in all our experiments at $I^{\max} = 0.625\,\mu W/\mu m^2$. The value of $I^{\max}$ sets the largest particle velocity to $v_p^{\max} = 2\,\mu m/s$. Figure 2b–d show experimental snapshots of a dilute suspension of APs prior (top) and after (bottom) interacting with a single light pulse traveling from left to right with different velocities. At low pulse speeds $u \ll v_p^{\max}$ all APs are displaced to the right, at $u \approx v_p^{\max}$ APs are transported in both directions (mostly to the left resembling the diffusing wave paradox) and for $u \gg v_p^{\max}$ no significant changes in the position of the particles are observed. To quantify the APs response to a traveling pulse, we calculated their averaged displacement $x = \frac{1}{N}\sum_{i=1}^{N} x_f^i - x_0^i$, where $x_0^i$ and $x_f^i$ are the initial and final positions of the $i$th particle prior and after interaction with the pulse. While at low pulse velocities, the particle displacement is along the pulse traveling direction, a motion counter to the pulse is observed for $u \approx v_p^{\max}$ (Fig. 2e). As will be shown later below, such behavior is in quantitative agreement with numerical simulations of APs exhibiting positive phototaxis. It should be mentioned that in absence of aligning torques the opposite behavior, i.e. displacement counter (along) the pulse direction at low (high) pulse velocities is observed[34,36]. The saturation of the displacement at small $u$ in Fig. 2e is due to our finite field of view, which provides an upper bound of $\Delta x = 100\,\mu m$. With increasing pulse width, the displacement counter to the pulse motion becomes weaker and eventually vanishes.

To understand the AP motion in more detail, we analyze their time-resolved positional and orientational response to a passing traveling pulse. At small pulse velocities $u/v_p^{\max} \ll 1$, the situation resembles the static case discussed above because the AP will almost instantaneously align parallel with intensity gradient $\nabla I$ of the slowly traveling pulse (Fig. 1). Since the pulse travels to the right, the positively phototactic APs spend more time in the back than in the front of the pulse, which causes an effective AP motion to the right (Fig. 3a). Since APs can easily catch up with the slow pulse, a similar oscillatory trajectory as shown in Fig. 1b with almost periodic particle reorientations is superimposed to the drift motion.

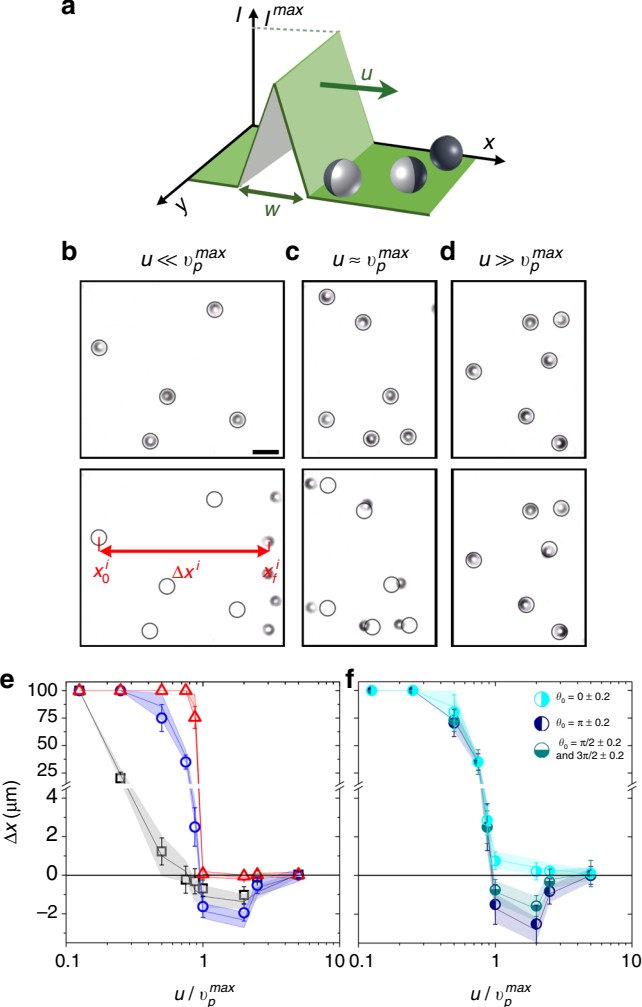

**Fig. 2** Light pulse traveling across phototactic active particles. **a** Sketch of a traveling light pulse with width $w$, maximum intensity $I^{\max}$ and traveling velocity $u$. **b**–**d** Experimental snapshots of particle configuration (top) before and (bottom) after interacting with a light pulse with $w = 4$ and for pulse velocities (**b**) $u \ll v_p^{\max}$, (**c**) $u \approx v_p^{\max}$, (**d**) $u \gg v_p^{\max}$. To enhance the visibility of particle displacements during the interaction with the pulse, black circles indicate their initial positions $x_0^i$. The red arrow indicates the particle displacement $x^i = x_f^i - x_0^i$ with $x_f^i$ their position after the pulse has swept over the corresponding particle. Scale bar is 5 μm. **e** Averaged particle displacement $x$ as a function of $u$ (in units of particle maximum velocity $v_p^{\max}$) for $w = 2.5\sigma$ (gray), $w = 4\sigma$ (blue), and $w = 7\sigma$ (red) obtained from experiments (symbols) and numerical simulations (shaded areas). **f** Same data as in (**e**) but now resolved depending on the initial AP orientation (see legend) for $w = 4\sigma$. Data average for at least 25 realizations with random initial particle orientations. The error bars represent the s.d

Within increasing pulse velocity particles spend less time within the pulse. For $u \approx v_p^{\max}$, the residence time $t^{res}$ of APs in the front and back of the pulse (horizontal arrows in Fig. 3b, c) are below 2 s each which is below the particle's reorientation time $\tau$. Therefore, APs have not enough time to align their orientation parallel to the light gradient of the traveling light pulse. As a consequence, their spatial displacement by a pulse becomes strongly dependent on their initial orientation $\theta_0$ (cf. Fig. 2c). Exemplarily, we show trajectories for two initial AP orientations, i.e. $\theta_0 \approx 0$ (Fig. 3b) and $\theta_0 \approx \pi$ (Fig. 3c), which demonstrate particle displacement along and counter to the pulse propagation

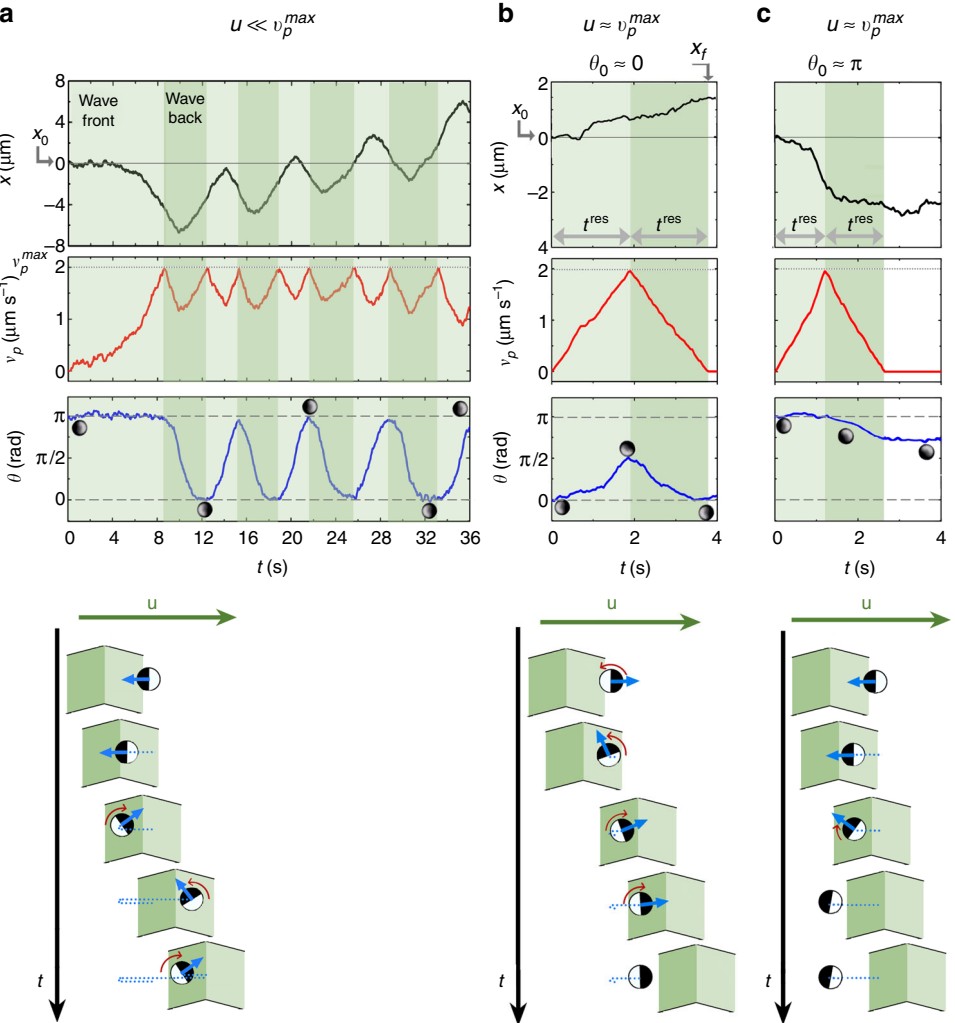

**Fig. 3** Time-resolved response of APs to a traveling symmetric light pulse. **a** $u \ll v_p^{max}, v_p^{max} = 2\,\mu m/s$, $u = 0.25\,\mu m/s$. **b** $u \approx v_p^{max}$, $v_p^{max} = 2\,\mu m/s$, $u = 2\,\mu m/s$. **c** $u \approx v_p^{max}$, $v_p^{max} = 2\,\mu m/s$, $u = 2\,\mu m/s$, $\theta_0 = \pi$. Temporal evolution of $x$, $v_p^{max}$, and $\theta$ and schematic representation of the AP's response to the traveling pulse (top view). Background color code: Light and dark green areas represent wavefronts and wavebacks, respectively

direction. For a qualitative understanding, we provide a sketch at the bottom of Fig. 3b, c. For $\theta_0 \approx 0$, the particle initially points antiparallel to the gradient $\nabla I$ of the pulse front and will start to propel in the direction of pulse propagation. Any deviation from $\theta_0 = 0$ (e.g. due to thermal noise) causes a torque acting on the particle (Eq. (1)). Since $t^{res} < \tau$, parallel alignment between **n** and $\nabla I$ is not achieved; instead, particles tumble between $\theta_0 \approx 0$ and $\pi/2$. This tumbling motion is at the expense of the translational particle drift which is significantly smaller compared to Fig. 3a. On the opposite, particles having an initial orientation $\theta_0 \approx \pi$ are aligned parallel to $\nabla I$ and propel counter to the pulse until reaching the intensity maximum (Fig. 3c). The torque $M$ is stabilizing AP orientations $\theta_0 \approx \pi$ (Eq. (1)), therefore particles maintain their orientation while propelling towards the pulse maximum. Once the AP is in the back of the pulse, **n** becomes antiparallel to $\nabla I$ resulting in particle reorientation. Since $t^{res} < \tau$, however, the particle essentially maintains its original alignment counter to the pulse traveling direction which then leads to the diffusing wave paradox (Supplementary Movie 2). The influence of the initial particle orientation $\theta_0$ to their response to a traveling pulse is also seen in Fig. 2f where we have replotted the data of Fig. 2e but now resolved according to $\theta_0$.

Finally, when $u/v_p^{max} > 1$, the particles' residence time become further reduced, which leads to a strongly weakened phototactic

response and thus to a decreasing displacement an agreement with Fig. 2e.

The observed disappearance of the diffusing wave paradox with increasing pulse width (Fig. 2e) is easily understood by considering that $t^{res}$ increases with $w$. For larger $w$ particle alignment parallel to $\nabla I$ becomes more likely and thus facilitates particle motion along the pulse traveling direction.

As shown in Fig. 3b, c the particle orientation before and after interacting with a light pulse with $\theta_0 \approx 0$ hardly changes (note that considerable reorientations occur when the pulse is running over the particle). Therefore, when subjecting APs to a train of $N$ identical pulses with period $T$ (Fig. 4a), at first glance one expects that the displacement is enhancement by a factor $N$. Figure 4b shows the averaged particle displacement $\Delta x^{N=10}(N = 10)$ with $u/v_p^{max} = 1$, as a function of $T/D_r$, i.e. time $T$ in units of the AP's rotational diffusion time. Again, we have independently analyzed the data for different initial AP orientations (before encountering the first pulse) and exemplarily show the results for $\theta_0 = 0 \pm 0.2$ and $\pi \pm 0.2$. For small $T$, indeed the displacements are about 10 times larger than for a single pulse (Fig. 2f and Supplementary Movie 3). With increasing $T$, however, the total displacement decreases and almost disappears. This is caused by the particles' rotational diffusion time $1/D_R$, which determines their orientational dynamics between the pulses and

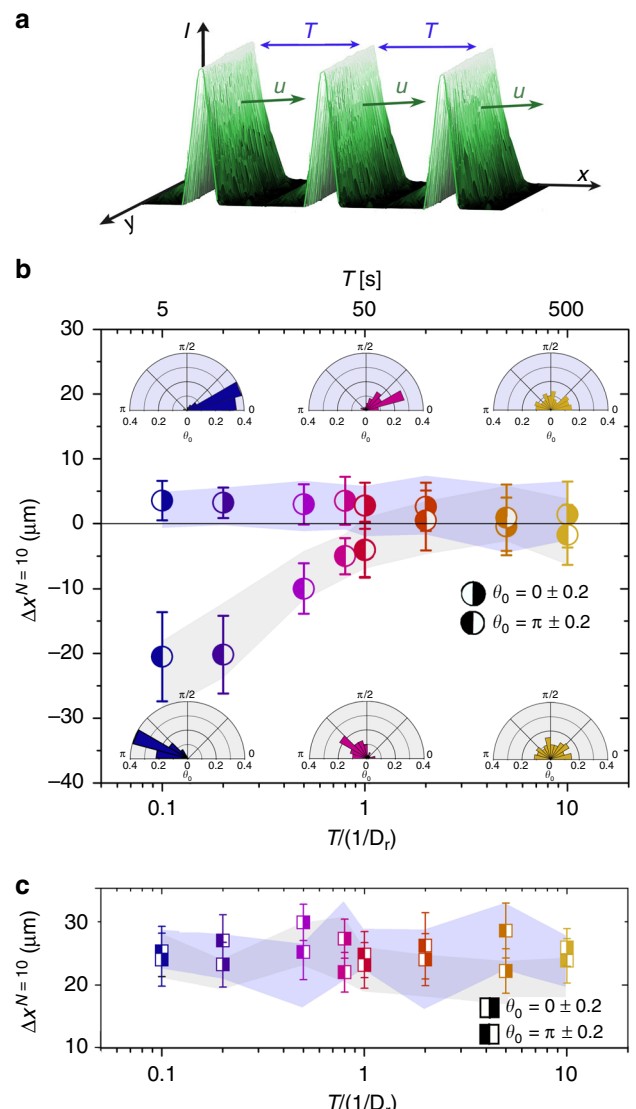

**Fig. 4** Response of APs to periodic pulse trains. **a** Measured intensity profile of a train of light pulses with $T$ the time intervals between the pulses and $u$ the pulse velocity. **b** Total AP displacement $\Delta x^{N=10}$ after interaction with 10 pulses with $u/v_p^{max} = 1$ and $w = 4\sigma$. Symbols (experimental data) correspond to different initial AP orientations $\theta_0$ (see legend) and numerical simulations (shaded areas). The insets show the probability distributions of the particle orientation prior to each pulse within the train for $T/(1/D_r) = 0.1$ (blue), 0.6 (pink), 10 (yellow). **c** Corresponding $\Delta x^{N=10}$ for $u/v_p^{max} = 0.875$ for $w = 4\sigma$. Symbols (experimental data) correspond to different initial AP orientations $\theta_0$ and numerical simulations (shaded areas). All experimental data averaged over at least 20 realizations each

leads to an increasing randomization, i.e. decorrelation of the particle orientation with increasing $T$. The orientational decorrelation between single pulses is shown as insets in Fig. 4b. While the particle orientation prior to each pulse remains rather preserved at small $T$, particles are almost isotropically aligned when $T/(1/D_r) \ll 1$ and, thus, explains why eventually no net particle transport is achieved. With decreasing pulse velocity, the time APs spend within a pulse finally exceeds their reorientation time. Accordingly, they will perfectly (parallel) align to the local gradient of pulses and the displacement becomes independent of the time interval of pulses. This is shown in Fig. 4c for $u/v_p^{max} = 0.875$ and demonstrates that

transport of active particle by pulse trains is rather robust regarding parameter variations.

**Sorting mechanism.** Apart from steering phototactic particles by trains of light pulses, this approach can be also employed for sorting APs. Since torques subjected by light gradients increase with the particle diameter $\sigma$[27,40], the AP's response to laser pulses becomes size-dependent. Exemplarily we discuss this for a binary mixture of APs with $\sigma^{small} = 3.25\,\mu m$ (rotational diffusion time $1/D_r^{small} \sim 50\,s$ and reorientation time $\sigma^{small} = 3.3 \pm 0.75\,s$) and $\sigma^{big} = 4.9\,\mu m$ ($1/D_r^{big} \sim 115\,s$ and $\tau^{big} = 1.83 \pm 0.55\,s$) subjected to a periodic pulse train. It should be noted that the propulsion velocity only depends on the cap thickness but not on $\sigma$[37]. Figure 5a shows typical trajectories of big (red) and small (blue) APs, relative to their initial position $x_0$ at $t = 0\,s$. The data were obtained for $u = 2\,\mu m/s$ ($u/v_p^{max} = 1$) and $T = 5\,s$. As a result of the faster orientational response of the large APs (red trajectories), they can follow the propagating wave (thereby performing a similar back-and-forth motion as shown in Fig. 3a) and move to the right. On the opposite, small particles (blue trajectories) travel on average counter to the pulse train. Note that also motion along the pulse train becomes possible when $\theta \approx 0$, but then with much smaller displacements compared to $\theta \approx \pi$ (compare Fig. 4b). Figure 5b shows the corresponding time-dependent probability distributions of small and large APs. With increasing time, we observe a broadening but also a shift of the mean values in opposite directions (Supplementary Movie 4).

**Discussion**
In addition to our experiments, we also performed numerical simulations where we considered that (i) the AP propulsion velocity linearly increases with the local light intensity (Fig. 1b) and that (ii) an additional angular velocity is imposed on the APs in the presence of a light gradient (Eq. (2)). Since the light pulses travel along to the $x$-direction, the propulsion velocity and the orientational dynamics depend on space and time (the orientational dynamics additionally depends on the particle orientation $\phi$ relative to the local light gradient). For sake of simplicity, we approximated the intensity profile of a single pulse by two identical segments with constant and opposite gradients $\nabla I$. Thus, the vectorial translational and rotational Langevin equations can be written as

$$\dot{\mathbf{r}} = v_p(x)\mathbf{n} + \boldsymbol{\xi}_\mathbf{r}, \quad (3)$$

$$\dot{\boldsymbol{\theta}} = \omega_{max}\sin\phi(x,t) + \boldsymbol{\xi}_\theta, \quad (4)$$

where $\theta$ is the angle between the $x$-axis and particle orientation (inset Fig. 1b). The prefactor $\omega_{max}$ has been obtained from the theoretical fit in Fig. 6b (see the section "Methods" for further details). Brownian fluctuations are included by means of zero-mean Gaussian noise terms $\boldsymbol{\xi}_\mathbf{r}$ and $\boldsymbol{\xi}_\varphi$ defined by the variances $\langle \boldsymbol{\xi}_\mathbf{r}(t_1) \otimes \boldsymbol{\xi}_\mathbf{r}(t_2) \rangle = 2D_{tr}\mathbf{1}\delta(t_1 - t_2)$) and $\langle \boldsymbol{\xi}_\theta(t_1)\boldsymbol{\xi}_\theta(t_2) \rangle = 2D_r\delta(t_1 - t_2)$), where $\otimes$ denotes the dyadic product, $\mathbf{1}$ is the unit tensor, and $D_t$ and $D_r$ are the translational and rotational diffusion coefficients of a spherical Janus particle, respectively. Our numerical results (cf. Figs. 2 and 4) show excellent agreement with our experimental data and confirm that a spatio-temporal variation of the propulsion velocity in combination of an aligning torque is sufficient to understand particle motion along and counter to the direction of a traveling light pulse. Contrary to experimental trajectories which are limited by the field of view, we obtained simulated trajectories with total lengths up to 2000 μm. This allowed us also, to investigate the behavior of APs in pulse trains at much larger particle propulsion velocities compared to our experiments. In Fig. 6a, we show

computed trajectories $x(t)$ for different $v_p^{max}$. For $u \ll v_p^{max}$, the pulse is traveling too fast, so the particle comes almost immediately to rest when hitting the pulse, i.e. no net transport. In the optimal case, $v_p^{max}$ slightly larger than $u$, the particles are able to be reoriented within the pulse, which increases dramatically the directed particle motion. When $v_p^{max} \gg u$, the particles overcome the pulse without being reoriented. The long steady-state

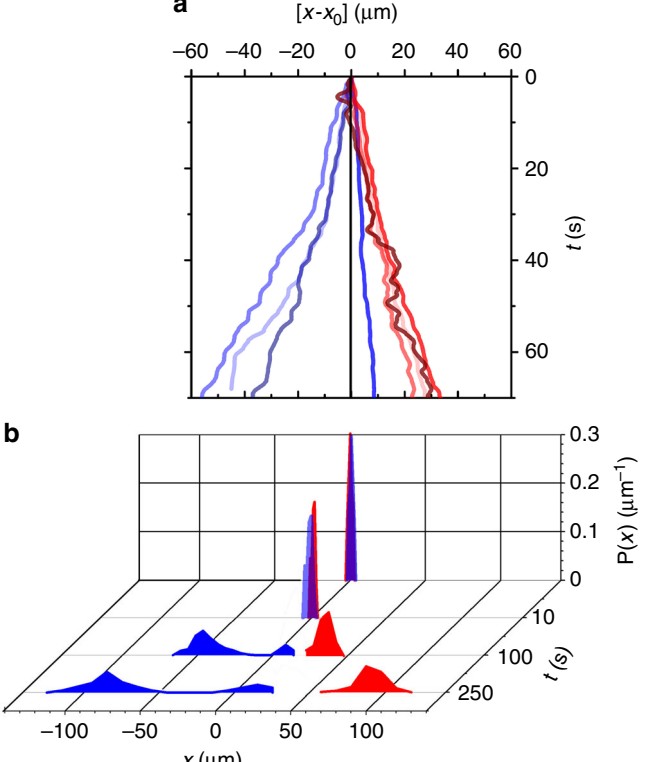

**Fig. 5** Sorting of a binary suspension of APs by periodic trains of traveling pulses. **a** Experimental trajectories for different particle diameters $\sigma = 3.25$ μm (blue) and 4.9 μm (red) for $u = 2$ μm/s, $v_p^{max} = 2$ μm/s, $w = 4\sigma$ and $T = 5$ s. **b** The corresponding time evolution of the probability distribution $P(x)$ of each active species $\sigma = 3.25$ μm (blue) and 4.9 μm (red). Data averaged over at least 30 trajectories each

trajectories enable to extract the AP's mean velocity of the $V$ by doing a linear fit (see inset Fig. 6a), which is shown in Fig. 6b. As expected $V$ increase when $v_p^{max} \simeq u$, that is when the mean residence time in the pulse becomes larger. Consequently, this protocol offers promising opportunities for sorting.

Our results show a complex response of APs to traveling light pulses. Contrary to amoebae, where a similar response results from the organisms capability to integrate and adapt to external signals[6,7], APs respond strictly local and do not display memory. Instead, the APs response is due to the interplay of a modulation of their propulsion velocity and their phototactic properties. Because changes in the AP's propulsion direction require their rotation which is limited by viscous friction with the solvent, this enables their motion towards or opposite to the direction of the pulse depending on the pulse velocity. This allows for a novel bidirectional steering strategy of APs which is contrast, to e.g. topographical structures[42,43] or static optical landscapes[12,27,44–46], where only unidirectional particle motion is observed. Apart from particle steering, the use of traveling pulses may be employed for effective sorting of APs according to their size, shape but also swimming velocity. Finally, it should be pointed out that the observed behavior does not only apply to the specific propulsion mechanism discussed here, but is also applicable to APs powered by chemotactic[23,26,47], catalytic[48–51], or thermophoretic[52–54] forces.

## Methods

**Fabrication of APs and quantification of phototactic response.** Active colloids were made from spherical silica particles (diameter $\sigma = 3.25$ μm) half-coated with 50 nm carbon caps. As solvent, we used a binary critical mixture of water–2,6-lutidine having a lower critical point at $T_c = 34.1$ °C. The entire sample cell is kept at a temperature of $T_0 = 31$ °C using a bath cryostat. When the particles are homogeneously illuminated, the carbon cap becomes evenly heated by absorption of the laser light and its temperature will increase. When the cap's temperature exceeds $T_c$, the solvent near the cap will demix resulting in the appearance of a droplet nucleating around the particle. As a result of local body forces and the wetting properties of the particles, this leads to a self-propelling particle motion. At low illumination intensities $I < I_r$ the particle moves with the cap in the rear but reverses its direction of motion for $I > I_r$[37] (Fig. 7a). Independent of the intensity, the AP performs a persistent random walk with a transition from a short-time ballistic to a long-time effective diffusive behavior[38]. Contrary to homogeneous illumination, the AP motion is no longer isotropic in the presence of a light gradient $\nabla I$. Under such conditions, the cap becomes unevenly heated leading to the nucleation of asymmetric droplets. This results in a torque which aligns the cap always (independent of $I$) towards larger intensities, i.e. the particle orientation **n** becomes parallel to $\nabla I$[37]. This eventually leads to a negative and positive phototactic behaviors for $I < I_r$ and $I > I_r$, respectively (Fig. 7a). All experiments in this

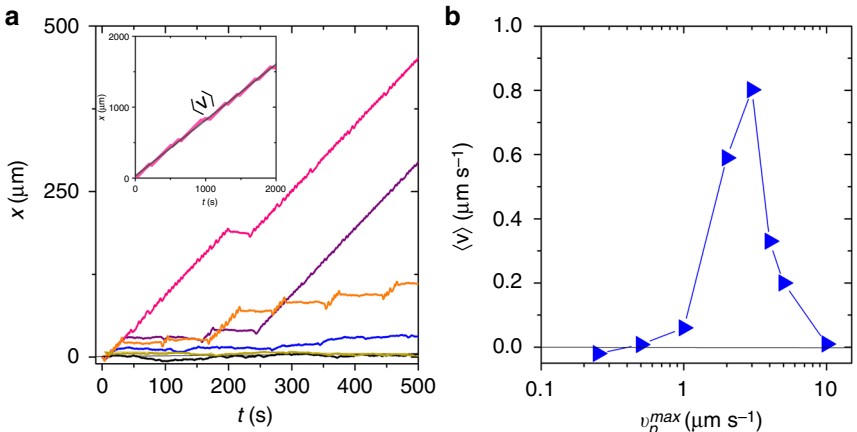

**Fig. 6** Transport of APs with different velocities by periodic trains of traveling pulses. **a** Numerical trajectories for APs with different propulsion velocities interacting with a train of pulses for $u = 1$ μm/s, $w = 4\sigma$, and $T = 50$ s ($v_p^{max} = 0.5$ μm/s (black), $v_p^{max} = 1.0$ μm/s (blue), $v_p^{max} = 2$ μm/s (purple), $v_p^{max} = 3$ μm/s (pink), $v_p^{max} = 5$ μm/s (orange), $v_p^{max} = 10$ μm/s (yellow)). (Inset) Long-time particle trajectory ($v_p^{max} = 3$ μm/s) with the solid line corresponding to a linear fit with slope $\langle V \rangle$. **b** Corresponding $\langle V \rangle$ vs. $v_p^{max}$

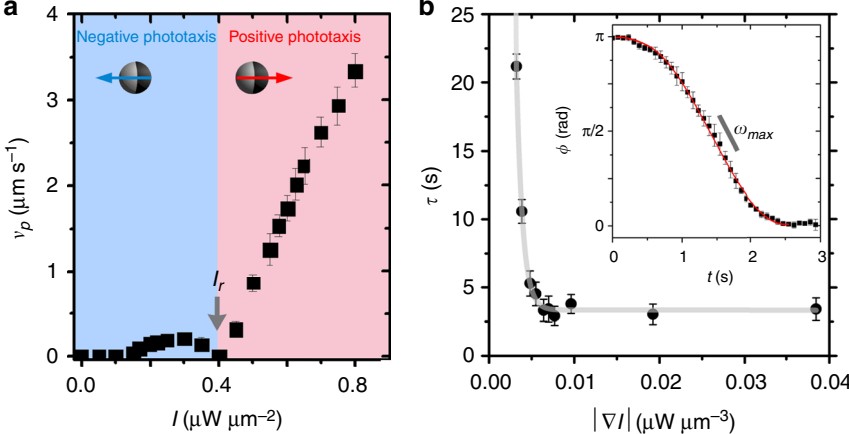

**Fig. 7** Light activated motion. **a** Experimentally measured propulsion velocity $v_p$ versus illumination intensity . Below and above $I_r$, particle move with the cap in the rear and the front. In the presence of a light gradient, additional torques are acting on the particle, which leads to negative and positive phototactic motions below and above $I_r$, respectively[37]. **b** Reorientation time $\tau$ as a function of the gradient magnitude $|\nabla I|$. The solid curve shows a theoretical fit (see the section "Methods" for details). The error bars represent the s.d. (Inset) Time evolution of the angle for an AP within a light gradient $|\nabla I| = 0.018\ \mu W\ \mu m^{-3}$. The solid curve shows the theoretical fit (see the section "Methods" for details)

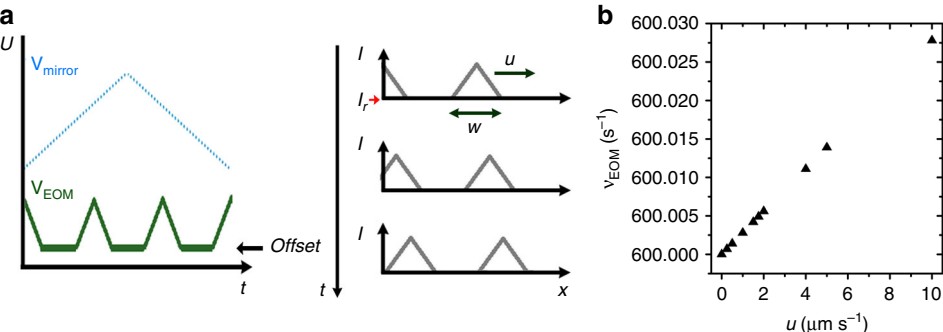

**Fig. 8** Experimental realization of traveling light pulses. **a** Schematic drawing of the voltages applied to the mirror and the EOM (left) and the resulting time-dependent traveling laser pulse with propulsion velocity, and width $w$ (right). **b** Dependence of the $\nu_{EOM}$ on the pulse velocity $u$ for $\nu_{mirror} = 200$ Hz

paper have been performed for intensities larger than $I_r$, i.e. under condition where the AP exhibits positive phototactic behavior, and where the propulsion velocity almost linearly increases with $I$ (Fig. 7).

To quantify the orientational response of particles to light gradients, which is relevant for the AP's encounter within the light pulse, we have measured their orientational response $\phi(t)$, i.e. the temporal change of the angle between $\nabla I$ and the particle orientation $\mathbf{n}$, when subjected to a light gradient. The inset of Fig. 7b shows the corresponding data for a particle with initial orientation $\phi = \pi$ and how it aligns parallel with $\nabla I$ within ~2 s (in our experiments clockwise and anticlockwise rotation is observed). The AP's reorientation dynamics in an intensity gradient $\nabla I$ is (in absence of noise) described by the differential equation (2). Solving this equation gives $\cos\phi(t) = \tanh(\omega_{max}(\bar{t} - t))$, where $t$ is the time when $\phi(\bar{t}) = \pi/2$ and $\omega_{max}$ as the only fitting parameter used to obtain the theoretical fit in Fig. 7b and which strongly depends on $\nabla I$[27]. From this, one obtains the reorientation time $\tau = \frac{2}{\omega_{max}}\ln\left(\frac{\cos(\phi_{max})+1}{\sin(\phi_{max})}\right)$, where $\phi_{max}$ is the total rotation[27]. The reorientation times given in this paper correspond to the value for $\phi_{max} = 3$ rad, which is shown in Fig. 7b vs. $\nabla I$. As a result of the heat flux through the particle and its coupling to the surrounding solvent velocity at the particle surface, $\tau$ saturates at large gradients[27]. All gradients considered in this work ($0.007\ \mu W\ \mu m^{-3} < |\nabla I| < 0.04\ \mu W\ \mu m^{-3}$) are above the saturation value and yield a constant reorientation time $\tau = 3.3 \pm 0.75$ s.

Creation of dynamical light patterns is achieved by a periodically oscillating mirror, which scans the elongated focus of a line-shaped (1 μm × 2000 μm) laser beam ($\lambda = 532$ nm) across the sample plane. For small voltages applied to the mirror ($V_{mirror}$), the displacement of the laser line is proportional to $V_{mirror}$. Synchronization of the scanning motion with the input voltage of an electro-optical modulator ($V_{EOM}$) leads to spatio-temporal illumination patterns[27]. To generate a static light pattern, the ratio of the mirror-frequency $\nu_{mirror}$ and the EOM-frequency $\nu_{EOM}$ must yield a rational number, which determines the number of intensity maxima in the field of view. In order to create time-dependent illumination patterns, this ratio must be slightly varied (in our experiments we kept $\nu_{mirror}$ constant at 200 Hz and varied $\nu_{EOM}$). This is schematically shown in Fig. 8a for the situation, where $\nu_{mirror}$ and $\nu_{EOM}$ are modulated according to a symmetric triangular and truncated triangular function, respectively. The time interval where $V_{EOM}\ \phi_{max} \neq$ constant determines the width of the resulting traveling pulses. Figure 8b shows the dependence of the pulse velocity $u$ on $\nu_{EOM}$ which increases linearly for small deviations from a rational number of the frequencies of the mirror and the EOM.

Due to the large thermal diffusivity of the carbon cap (~$10^{-7}$ m² s$^{-1}$), the temperature and mixture's concentration profile around the particle will respond on times scales below $10^{-5}$ s to changes in the illumination intensity[37]. For the given laser scanning frequencies, this leads to quasi-static illumination conditions and an immediate response of the particle propulsion to the spatio-temporal light field.

**Particle tracking.** Images of the particles were acquired using video microscopy with a frame rate of 12 fps. The size of the region of interest is 300 μm × 160 μm. Particle tracking was performed using an automated tracking program developed in-house with Matlab image analysis software[55]. First, the background of images was removed by using an appropriate intensity threshold. The particle position $\mathbf{r} = (x,y)$ was approximated as the center of mass of the contours obtained after segmentation, with a spatial resolution ~100 nm. Particle trajectories were calculated using Bayesian decision-making, linking every particle center with the previous closest one. Because of the optical contrast between the dark carbon hemisphere and the transparent silica, the particle orientation vector $\mathbf{n} = (\cos\theta, \sin\theta)$ can be obtained from the vector connecting the particle center and the intensity centroid of the particle image. The error of this detection is <5% as confirmed by comparison with stuck particles whose orientation can be precisely controlled.

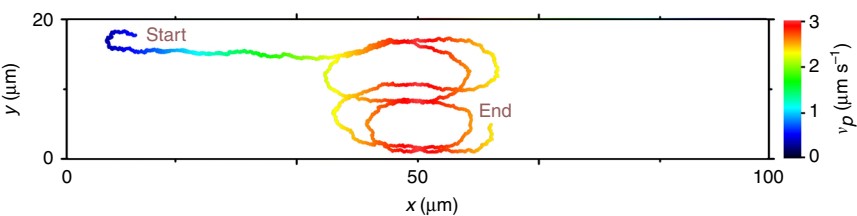

**Fig. 9** Numerically calculated trajectory of an AP in a static intensity profile. Example of a numerical trajectory in a static triangular velocity profile (same profile as in Fig. 1) with $v_p$ labeled in color

**Simulated particle trajectories in a static motility gradient**. To demonstrate, that the AP dynamics in light gradients are correctly described by our numerical simulations, we compared the experimentally observed dynamics of an AP in a static triangular motility profile (cf. Fig. 1). Indeed, the simulations show very similar trajectories, in particular, the oscillatory motion near the intensity maximum is reproduced very well. The results show good agreement between the point-like particles simulations (Fig. 9) and the experimental measurements (Fig. 1). For each experimental conditions, between 25 and 50 simulations were done.

## Data availability

The data that support the findings of this study are available from the corresponding author upon request.

## Code availability

The custom codes used in this study are available from the corresponding author upon request.

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

## Acknowledgements

C.B. acknowledges financial support by the ERC Advanced Grant ASCIR (Grant no. 693683) and from the German Research Foundation (DFG) through the priority program SPP 1726.

## Author contributions

C.L. and C.B. designed the research, discussed the data, and wrote the paper. C.L. carried out the experiments, simulations, and analyzed the data.

## Additional information

**Competing interests:** The authors declare no competing interests.

