## [Peer Review File · Nature Communications]

Reviewers' comments:

Reviewer #1 (Remarks to the Author):

The paper entitled "Diffusing Wave Paradox of Phototactic Particles in Traveling Light Pulses" by C. Lozano & C. Bechinger reports an experimental study of phototactic particles in presence of shifting light patterns. This work demonstrate that light-sensitive Janus colloids move mostly along the same direction of the travelling light pattern.

However for certain values of the pattern speed the authors find a slow (yet significant) motion of the particles in the opposite direction. They point out that the key ingredient generating such a complex response is the reorientation of the colloids along the light intensity gradient. Since the orientational dynamics depends strongly on the colloids size it is shown that a travelling illumination pattern can separate active colloids of different radii. Finally it is shown that computer simulations of active particles (in a dynamic activity gradient) capture very well the experimental data.

This work is very interesting since it addresses a very hot topic at the moment, i.e. how to control structure and dynamics of active light-sensitive colloids and living cells suspensions. The findings reported are intriguing, the paper is well written and the discussion of the results is comprehensible also for inexperienced readers.

However the article, in its present form, has a couple of points which are not discussed in depth and that the authors should consider to improve the paper:

(1) At the end of the introduction they write: "In view of such limitations, it sounds surprising that numerical simulations of chemotactic APs suggest their motion counter to a traveling chemical concentration pulse [32,33]. So far, however, such theoretical predictions have not been confirmed in experiments." In my opinion this sentence is a bit misleading and should be revised:

-In both references [32] and [33] it is specifically mentioned that the models are built having in mind light-sensitive particles/cells (subjected to shifting light/heating patterns).

-It sounds like the present work will verify the theoretical predictions of [32,33]. However one of the main conclusions of these works was that, if light-induced reorientation is negligible, the particles tend to travel oppositely with respect to a slow light pattern, while travelling along the same direction of a fast pattern. In the present work the authors show the exact opposite: the particles shift in the same direction of a slow illumination pattern, while inverting swimming direction for fast patterns.

I believe this is an important qualitative difference (due to the dominant reorientation mechanism of Janus colloids) that the authors should discuss accurately and maybe cite a recent experimental work [a] which instead focuses on photosensitive bacteria which do not significantly reorient with respect to the gradient.

(2) The authors make a detailed introduction about the "diffusing wave paradox" in living organisms (in particular they take *Dictyostelium* as an example). From the discussion that follows it is however unclear how close is the observed behaviour with respect to the one of chemotactic cells. To which extent can the orientational response mimic the chemotactic behaviour of real biological systems? In some previous works [b] it was discussed that adaptation plays some role in determining the chemotactic behaviour in presence of traveling waves (i.e. the response to concentration of chemoattractant decreases at high concentrations). I would be interesting to stress the analogies and differences between the model used in the simulations by the authors and the one used in [b] to model "travelling wave chemotaxis" in *Dictyostelium*. Can the two models be mapped onto each other in some limits and/or under suitable approximations?

In conclusion I believe the paper is very interesting and should be published with a few minor improvements that would make it more focused and enrich the discussion.

Finally I have a few minor suggestions/questions:

- What are the three big green triangles in Fig.2(f)?
- Could Fig.2(e) and (f) (and also Fig.4(b) and (c)) be supplemented with a legend indicating what the different symbols represent?
- The authors could cite ref [c] alongside with [40]

[a] N. Koumakis et al., arXiv:1811.09133, (2019)

[b] R. E. Goldstein, Phys Rev Lett. 77, 775 (1996)

[c] G. Frangipane et al. eLife 7, e36608 (2018)

Reviewer #2 (Remarks to the Author):

Title: Diffusing wave paradox of phototactic particles in traveling light pulses

Authors: C. Lozano and C. Bechinger

The authors investigate the motion of Janus colloidal particles which responds to the external light illumination. In the previous paper by the authors, it was reported that the Janus colloid particles self-propell due to the effect of the heating related to the light emission. It was also reported that the Janus colloidal particle tends to move in the direction of local gradient of brightness by the reorientation of the graphite-covered part. In the present study, the authors report the response to the traveling light pulses. They experimentally investigated the Janus colloidal particle motion under the situation where the region with strong light illumination is propagating like a wave. This situation is motivated by the "diffusion wave paradox" which is well known in a system with *Dictyostelium*. They also proposed a simple model and reproduced the experimental results by numerical calculation.

It is interesting to address the "diffusion wave paradox" using Janus colloidal particles. However, I do not recommend the present manuscript for Nature Communication due to the following reason:

The authors previously published a paper "Phototaxis of synthetic microswimmers in optical landscapes" in Nature Communications in 2016 [Reference 25], and they reported the phototaxis of the Janus colloidal particles. They also discuss the mechanism using a model, and clarify the reorientation of the particles. Therefore, the novel point of the present manuscript is that the Janus colloidal particles show the "diffusing wave paradox". In the abstract, the authors write "Despite their entirely memory-less, i.e. strictly local response to the environment, we observe particle motion counter to the pulse direction." However, I consider that the orientation of the Janus colloidal particle is regarded as a "memory", since they experimentally measured the reorientation time. In the other paper (e.g. Ref 11), it was suggested that the "diffusion wave paradox" can occur if the cell has an internal memory. Whether the reorientation of the Janus colloidal particle is regarded as an internal memory or "response to the environment" is the subjective problem, and the main results of this manuscript are easily constructed from their previous results and the other papers on the "diffusion wave paradox".

The realization of the "diffusion wave paradox" in the Janus colloidal particles is interesting and may be worth publishing in other journal like Scientific Reports.

Reviewer #3 (Remarks to the Author):

This paper reports on experiments and simulations of laser-heated active particles in time-dependent intensity pulses. The work goes well beyond previous results on this system. The results are original and provide a nice example of complex dynamics of a rather simple system in a minimal patterned environment.

I have a few minor remarks on presentation.

1) The analogy with the chemotactic wave paradox of *Dictyostelium amoebae* is quite tempting. Still, the authors should make clear that the response of amoebae is more complex than that of their particles: *Dictyostelium* moves in upcoming cAMP wave but doesn't move in the back part of the signal, probably due to short-time memory.

2) I suppose the particle's backward move in each pulse is much shorter than that of *Dictyostelium*. For the latter, it's their velocity multiplied by the half pulse duration. The authors should compare this to the backward distance of their particles.

3) Also, I suppose the active particles need to point in the right direction whereas the amoebae detect the gradient.

In this respect, abstract and introduction are a bit misleading, it took me some time to single out the "paradox" for active particles. I suggest the authors point out the differences in the discussion.

4) The discussion of the particles' motile response in the Methods section is not clear to me. fig 7a shows a change of sign of the velocity with laser power, how is this related to the data of fig 4b?

On page 11 the reorientation time is derived from the power gradient, whereas the data of fig 4 are presented in terms of a rotational diffusion time.

It would be helpful if the authors presented first how the particle moves in a constant power, then discussed the features added by a gradient.

Thank you very much for your editorial efforts regarding our manuscript and for forwarding the referee reports to us.

Below, please find our point-by-point response to all their questions and remarks. To facilitate tracking all changes in the revised manuscript, we marked them in blue.

We believe that our manuscript has greatly benefited from the reviewers' feedback, and we hope that the revised version will be in a good position to be accepted by Nat. Comm.

Best regards,
Clemens and Celia

Reviewer #1

This work is very interesting since it addresses a very hot topic at the moment, i.e. how to control structure and dynamics of active light-sensitive colloids and living cells suspensions. The findings reported are intriguing, the paper is well written and the discussion of the results is comprehensible also for inexpert readers.

However the article, in its present form, has a couple of points which are not discussed in depth and that the authors should consider to improve the paper:

1. At the end of the introduction they write: "In view of such limitations, it sounds surprising that numerical simulations of chemotactic APs suggest their motion counter to a traveling chemical concentration pulse [32,33]. So far, however, such theoretical predictions have not been confirmed in experiments." In my opinion this sentence is a bit misleading and should be revised:

-In both references [32] and [33] it is specifically mentioned that the models are built having in mind light-sensitive particles/cells (subjected to shifting light/heating patterns).

-It sounds like the present work will verify the theoretical predictions of [32,33]. However one of the main conclusions of these works was that, if light-induced reorientation is negligible, the particles tend to travel oppositely with respect to a slow light pattern, while travelling along the same direction of a fast pattern. In the present work the authors show the exact opposite: the particles shift in the same direction of a slow illumination pattern, while inverting swimming direction for fast patterns. I believe this is an important qualitative difference (due to the dominant reorientation mechanism of Janus colloids) that the authors should discuss accurately and maybe cite a recent experimental work [a] which instead focuses on photosensitive bacteria which do not significantly reorient with respect to the gradient.

Response: The referee is absolutely right by noticing that the presence of an aligning torque is reversing the direction of motion of particles at slow and high pulse velocities compared to the torque-free case (see numerical studies of Geiseler et al.

and Maggi et al). In all cases, however, particle motion counter to the pulse direction (i.e. diffusing wave paradox) can be observed even without invoking memory or adaption effects. Since we wanted to stress the mere possibility of such counter-pulse motion in synthetic systems in the introduction, we found the above references at the beginning of the paper appropriate.

However, to avoid the possible impression that our work is a direct verification of these numerical studies, we have modified the corresponding part of the introduction and say now only, that “ it is surprising that numerical simulations suggest the principle possibility of AP motion against and along a traveling pulse^{34–36} “. We think, that this will avoid the impression, that our work intends to be a direct confirmation of those studies. In addition, we have deleted the sentence “So far, however, such theoretical predictions have not been confirmed in experiments”.

On page 5 (in the context of Fig.2e) we have added a brief discussion how the presence or absence of aligning torques reverses the direction of the particle displacement to a slow and fast propagating wave.

We are particularly thankful for making us aware of the preprint of Koumakis which demonstrates current reversal of E. Coli in traveling optical pulses at high and low pulse velocities. In fact, this corresponds exactly to the torque-free case in agreement with Geisler et al. PRE (2016). We have added the E. Coli reference as another example of the occurrence of the diffusing wave paradox in living systems in the introduction.

Changes:

Changed sentence in the introduction where we refer to numerical simulations of other authors: “Given these limitations compared to living systems, it is surprising that numerical simulations suggest the principle possibility of AP motion against and along a traveling pulse^{34–36}. “

We have added the preprint of Koumakis in the intro right after discussing *Dictyostelium* and added the paper of Geisler et al. Sci. Rep. 2017.

On page 5 we added a brief discussion on the importance of positive phototaxis to understand the direction of AP motion at low and high pulse velocities: “While at low pulse velocities u , the particle displacement is along the pulse traveling direction, a motion counter to the pulse is observed for $u \approx v_p^{max}$ (Fig. 2e). As will be shown later below, such behavior is in quantitative agreement with numerical simulations of active particles exhibiting positive phototaxis. It should be mentioned that in absence of aligning torques the opposite behavior, i.e. displacement counter (along) the pulse direction at low (high) pulse velocities is observed^{34,36}. “

2. The authors make a detailed introduction about the "diffusing wave paradox" in living organisms (in particular they take *Dictyostelium* as an example). From the discussion that

follows it is however unclear how close is the observed behaviour with respect to the one of chemotactic cells. To which extent can the orientational response mimic the chemotactic behaviour of real biological systems? In some previous works [b] it was discussed that adaptation plays some role in determining the chemotactic behaviour in presence of traveling waves (i.e. the response to concentration of chemoattractant decreases at high concentrations). I would be interesting to stress the analogies and differences between the model used in the simulations by the authors and the one used in [b] to model "travelling wave chemotaxis" in Dictyostelium. Can the two models be mapped onto each other in some limits and/or under suitable approximations?

Response: As already mentioned at the beginning of the paper, simple synthetic active particles do not exhibit the complexity of living systems (see also our response to referee 3). This is why we have written in our introduction (p.2). "Contrary to microorganisms, they are not equipped with complex internal signal pathways responsible to perform intricate signal processing including a time-delayed response to external stimuli but, instead, respond strictly local to their environment." Due to these fundamental differences, to our opinion a deeper comparison or even mapping between amoebae and APs makes no real sense (and perhaps may even a little bit misleading). So, their similarity is the phenomenological response to a travelling pulse but not the underlying mechanism. For example, adaptation behavior as discussed for amoebae in [b] but also in more detail in the paper from Höfler [13] would require a sizable time lag between particle the propulsion velocity and the laser illumination. This, however, is absent in our system which does not exhibit such a time delay.

To make the differences between living systems and synthetic APs more clear and to emphasize that we are fully aware that living systems display a much more complex behavior than synthetic active particles, we have modified the abstract and the introduction. In addition, we again mention this in the last paragraph of the paper where we summarize our findings and provide an outlook. We have also added some more information regarding the absence of a time-delayed response of APs to their environment in the Methods section where we discuss the propulsion mechanism.

Changes:

Abstract. Despite their entirely memory-less, i.e. strictly local response to the environment, we observe the same phenomenological behaviour, i.e. particle motion counter to the pulse direction.

Introduction. Contrary to microorganisms, the simple structure of APs does neither allow for intricate signal processing nor a time-delayed response to external stimuli³³. Instead, APs respond strictly local to their environment. Given these limitations compared to living systems, it is surprising that numerical simulations suggest the principle possibility of AP motion against and along a traveling pulse.

Summary (last paragraph of paper). Contrary to amoebae, where a similar response results from the organisms capability to integrate and adapt to external

signals^{6,7}, APs respond strictly local and do not display memory. Instead, the APs response is due to the interplay of a modulation of their propulsion velocity and their phototactic properties. Because changes in the AP's propulsion direction require their rotation which is limited by viscous friction with the solvent, this enables their motion towards or opposite to the direction of the pulse depending on the pulse velocity.

Methods. Due to the large thermal diffusivity of the carbon cap (approx. $10^{-7} \text{ m}^2 \text{ s}^{-1}$), the temperature and mixture's concentration field around the particle will respond on times below 10^{-5} s to changes of the illumination intensity. For the given laser scanning frequencies, this leads to quasi-static illumination conditions and an immediate response of the particle propulsion to the spatio-temporal light field.

I believe the paper is very interesting and should be published with a few minor improvements that would make it more focused and enrich the discussion.

Finally, I have a few minor suggestions/questions:

- What are the three big green triangles in Fig.2(f)?

Response: The green triangles (together with the small icons to their right) were meant as a sketch of (the side view) a wave approaching an AP to explain the different particle orientations relative to the wave. However, all symbols are clearly defined in the caption and we decided to delete this inset.

- Could Fig.2(e) and (f) (and also Fig.4(b) and (c)) be supplemented with a legend indicating what the different symbols represent?

Response: In fact, the symbols are explained in the figure caption. However, we agree that a legend might be better. In order to avoid the plots being overloaded, in the revised version we have added a legend for the symbols in Fig.2e and Fig.4b. In the caption of Fig.2f and 4c we added a remark, that the symbols are identical to Figs. 2e and 4b, respectively.

- The authors could cite ref [c] alongside with [40]

[a] N. Koumakis et al., arXiv:1811.09133, (2019)

[b] R. E. Goldstein, Phys Rev Lett. 77, 775 (1996)

[c] G. Frangipane et al. eLife 7, e36608 (2018)

Response: We have added the suggested references.

Reviewer #2 (Remarks to the Author):

The authors previously published a paper "Phototaxis of synthetic microswimmers in optical landscapes" in Nature Communications in 2016 [Reference 25], and they reported the phototaxis of the Janus colloidal particles. They also discuss the mechanism using a model, and clarify the reorientation of the particles. Therefore, the novel point of the present

manuscript is that the Janus colloidal particles show the "diffusing wave paradox". In the abstract, the authors write "Despite their entirely memory-less, i.e. strictly local response to the environment, we observe particle motion counter to the pulse direction." However, I consider that the orientation of the Janus colloidal particle is regarded as a "memory", since they experimentally measured the reorientation time. In the other paper (e.g. Ref 11), it was suggested that the "diffusion wave paradox" can occur if the cell has an internal memory. Whether the reorientation of the Janus colloidal particle is regarded as an internal memory or "response to the environment" is the subjective problem, and the main results of this manuscript are easily constructed from their previous results and the other papers on the "diffusion wave paradox".

The realization of the "diffusion wave paradox" in the Janus colloidal particles is interesting and may be worth publishing in other journal like Scientific Reports.

Response: The response of active particles to their environment is strictly local and prompt, i.e. their propulsive motion does not depend on the past. The same also applies to their active realignment in the presence of a light gradient. Even though the reorientation dynamics can be characterized by an orientation time, one must be careful with comparing this with the effect of memory. Once the light gradient is removed, the active particle reorientation will immediately stop (apart from rotational diffusion) due to the low Reynolds number (please note that our particles respond on timescales on the order of 10^{-5} s to changes in the light field (Methods section) (Gomez-Solano et. al Scientific Reports (2017))). Since this timescale is much shorter than any other relevant timescale in our system, no memory is observed on our timescale. We have added this information to the Methods section where we describe the details of our propulsion mechanism.

For the sake of completeness, we would like to point out that memory effects play an important role in the orientational dynamics of active particles when immersed in a viscoelastic fluid (Narinder et. al PRL (2018)). Such conditions, however, do not apply to our current study since our binary fluid exhibits entirely Newtonian behavior.

Compared to our previous paper on phototaxis, our current study is considerably different regarding the following aspects:

a) In our current manuscript, the particles display positive phototaxis (compared to negative phototaxis before). This is a very important aspect since it affects the direction of the particle displacement relative to the propagating light pulse. Only when the particles display positive phototaxis, the active reorientation will affect the direction of displacement depending on the pulse velocity. For negative phototaxis, the direction of particle displacement remains unchanged compared to the situation where no tactic response, i.e. no reorientation within light gradients occurs. We have clarified this on page 5 where we discuss Fig.2e. (see Changes below)

b) In our previous Nat.Comm. paper, we investigated particle transport in a static and asymmetric sawtooth-shaped optical landscape. Under such conditions, the direction of transport is entirely determined by the asymmetry of the static intensity pattern. In our current manuscript, we study a travelling and symmetric light pulse and find that

the direction of particle motion can be reversed by changing the pulse velocity. We do not immediately see that this outcome is obvious from our previous results but consider this as a novel observation which has also implications for other propulsion mechanisms (see summary).

c) Opposed to the situation of our previous Nat. Comm. paper (and the vast majority of experimental and theoretical publications on microswimmers) the APs are permanently active. This, however, is no longer the case in our current study where particles are only active for the short interval, they are illuminated by the running laser pulse. When exposed to a train of periodic pulses, this leads to an intermittent active-passive behavior. Under such conditions, the role of rotational diffusion becomes crucial since it leads to an increasing particle misalignment relative to their original orientation. As a result, the total particle displacement decreases with increasing time interval between consecutive pulses which results on opposite traveling directions for particles with different sizes, i.e. different rotational diffusion time. We are not aware of a similar observation in previous studies.

In summary, we believe that our present study on active particles in spatio-temporal light fields goes far beyond our previous work on phototaxis in static landscapes (as also explicitly stated by reviewer #3). We are confident that our work will be helpful for researchers working in physics, chemistry but also engineering and is thus suitable for the broad readership of Nature Communications.

Changes:

page 5: While at low pulse velocities u , the particle displacement is along the pulse traveling direction, a motion counter to the pulse is observed for $u \approx v_p^{max}$ (Fig. 2e). As will be shown later below, such behavior is in quantitative agreement with numerical simulations of active particles exhibiting positive phototaxis. It should be mentioned that in absence of aligning torques the opposite behavior, i.e. displacement counter (along) the pulse direction at low (high) pulse velocities is observed^{34,36}.

Methods Section: Due to the large thermal diffusivity of the carbon cap (approx. $10^{-7} \text{ m}^2 \text{ s}^{-1}$), the temperature and mixture's concentration profile around the particle will respond on times scales below 10^{-5} s to changes in the illumination intensity³⁷. For the given laser scanning frequencies, this leads to quasi-static illumination conditions and an immediate response of the particle propulsion to the spatio-temporal light field.

Reviewer #3

This paper reports on experiments and simulations of laser-heated active particles in time-dependent intensity pulses. The work goes well beyond previous results on this system. The results are original and provide a nice example of complex dynamics of a rather simple system in a minimal patterned environment.

I have a few minor remarks on presentation.

1. *The analogy with the chemotactic wave paradox of Dictyostelium amoebae is quite tempting. Still, the authors should make clear that the response of amoebae is more complex than that of their particles: Dictyostelium moves in upcoming cAMP wave but doesn't move in the back part of the signal, probably due to short-time memory.*

Response: We fully agree that living systems have a much more complex response than APs, in fact, this is one of our central messages. This is why we have written in our introduction (p.2). "Contrary to microorganisms, however, they are not equipped with complex internal signal pathways responsible to perform intricate signal processing including a time-delayed response to external stimuli but, instead, respond strictly local to their environment."

To make those differences more clear and to emphasize that we are fully aware that living systems display a much more complex behavior than synthetic active particles, we have modified the abstract and the introduction. In addition, we have mentioned this point once more in the last paragraph of the paper.

Changes:

Abstract. Despite their entirely memory-less, i.e. strictly local response to the environment, we observe the same phenomenological behaviour, i.e. particle motion counter to the pulse direction.

Introduction. In view of such differences to living systems, it sounds surprising that numerical simulations of chemotactic APs suggest a similar phenomenological behavior, i.e. their motion counter to a traveling chemical concentration pulse.

Last paragraph of paper. Contrary to amoebae, where a similar response results from the organisms capability to integrate and adapt to external signals^{6,7}, APs respond strictly local and do not display memory. Instead, the APs response is due to the interplay of a modulation of their propulsion velocity and their phototactic properties. Because changes in the AP's propulsion direction require their rotation which is limited by viscous friction with the solvent, this enables their motion towards or opposite to the direction of the pulse depending on the pulse velocity.

2. *I suppose the particle's backward move in each pulse is much shorter than that of Dictyostelium. For the latter, it's their velocity multiplied by the half pulse duration. The authors should compare this to the backward distance of their particles.*

Response: Contrary to Dictyostelium, whose response to the front and back of a traveling pulse is different (adaption), this is not true for active particles which are lacking such sophisticated behavior. The reason why both systems display similar

phenomenological behavior is that APs have to turn in order to change their direction of travel. Such turning events are not present in Dictyostelium (see also next point).

3. Also, I suppose the active particles need to point in the right direction whereas the amoebae detect the gradient.

Response: Yes, this is absolutely true (see above). Our particles have a constant polarity (due to the carbon cap) and will always swim in the direction of the cap (i.e. they have to point in the "right" direction). Accordingly, directional changes require the active reorientation of the entire particle (aligning torques) or rotational diffusion.

In this respect, abstract and introduction are a bit misleading, it took me some time to single out the "paradox" for active particles. I suggest the authors point out the differences in the discussion.

Response: We have clarified now in the abstract and the introduction that the similarity with amoebae is only related to their translational response to travelling pulses, i.e. amoebae and APs, are able to move counter a propagating wave. However, the underlying biological/physical mechanisms are entirely different. In particular, no body reorientation is required for an amoeba to change its direction of motion.

Changes: see point 1 of referee#3

4. The discussion of the particles' motile response in the Methods section is not clear to me. fig 7a shows a change of sign of the velocity with laser power, how is this related to the data of fig 4b?

Response: We agree that the discussion of the particle's propulsion was not perfectly clear and we have therefore rewritten the corresponding part in the Methods section and also changed Fig.7a. In addition, we also emphasized, both in the Methods section and the main text of the paper that all our measurements were performed in the regime where they exhibit a positive phototactic response with the capped side of the AP moving ahead.

Changes:

- a) We have rewritten the first paragraph of the Results-section where we discuss the propulsive particle motion
- b) We have also rewritten the first section of the Methods part (Fabrication of active particles and quantification of phototactic response)
- c) Revised Fig.7a

On page 11 the reorientation time is derived from the power gradient, whereas the data of fig 4 are presented in terms of a rotational diffusion time.

Response: In Fig.7, we show the relationship between the reorientation time (i.e. active torque) τ of APs in a light gradient $|\nabla I|$. This reorientation time is the relevant quantity for the reorientation of the AP *inside a light pulse*.

In Fig. 4, we discuss the response of APs to a series of light pulses which are separated by time T . During T , i.e. in the time interval between pulses, particles can change their orientation only due to their rotational diffusion D_R . This leads to a decorrelation of their angular orientation which then explains why the AP displacement decreases with increasing T . We have clarified this point when discussing Fig.4.

Changes:

This is caused by the particles' rotational diffusion time $1/D_R$ which determines their orientational dynamics between the pulses and leads to an increasing randomization, i.e. decorrelation of the particle orientation with increasing T .

Methods. To quantify the orientational response of particles to light gradients, which is relevant for the AP's encounter within the light pulse, we have measured their orientational response $\phi(t)$, i.e. the temporal change of the angle between ∇I and the particle orientation \mathbf{n} , when subjected to a light gradient.

It would be helpful if the authors presented first how the particle moves in a constant power, then discussed the features added by a gradient.

Response: In the Methods section (Fabrication of active particles and quantification of phototactic response), we have now first discussed the motion of APs under constant power, i.e. homogeneous illumination. Afterwards, the differences in a light gradient are discussed.

Changes: Revised first paragraph of the Methods section.

REVIEWERS' COMMENTS:

Reviewer #1 (Remarks to the Author):

The authors have answered all my questions. I believe the manuscript is now ready for publication.

Reviewer #2 (Remarks to the Author):

The authors have replied to my previous comments, and they have clarified the novel point compared with their previous paper. Now I understand the novel points and now recommend for the publication in Nature Communications as it is.

Reviewer #3 (Remarks to the Author):

The authors have replied to most of the comments and they have clarified several aspects.

Thank you very much for your editorial efforts regarding our manuscript and for forwarding the referee reports to us. There were no further comments of the referees, they support publication in its present form.

Reviewer #1: The authors have answered all my questions. I believe the manuscript is now ready for publication.

Reviewer #2: The authors have replied to my previous comments, and they have clarified the novel point compared with their previous paper. Now I understand the novel points and now recommend for the publication in Nature Communications as it is.

Reviewer #3: The authors have replied to most of the comments and they have clarified several aspects.

Best regards

Clemens